# Remdesivir or Nirmatrelvir/Ritonavir Therapy for Omicron SARS-CoV-2 Infection in Hematological Patients and Cell Therapy Recipients

**DOI:** 10.3390/v15102066

**Published:** 2023-10-08

**Authors:** José Luis Piñana, Inmaculada Heras, Tommaso Francesco Aiello, Irene García-Cadenas, Lourdes Vazquez, Javier Lopez-Jimenez, Pedro Chorão, Cristina Aroca, Carolina García-Vidal, Ignacio Arroyo, Eva Soler-Espejo, Lucia López-Corral, Alejandro Avendaño-Pita, Anna Arrufat, Valentín Garcia-Gutierrez, Elena Arellano, Lorena Hernández-Medina, Clara González-Santillana, Julia Morell, José Ángel Hernández-Rivas, Paula Rodriguez-Galvez, Mireia Mico-Cerdá, Manuel Guerreiro, Diana Campos, David Navarro, Ángel Cedillo, Rodrigo Martino, Carlos Solano

**Affiliations:** 1Hematology Department, Hospital Clínico Universitario, 46017 Valencia, Spain; nachoarroyomartin@gmail.com (I.A.); julmor96@gmail.com (J.M.); paularodriguezgv@gmail.com (P.R.-G.); micocerda@hotmail.com (M.M.-C.); dcampos@incliva.es (D.C.); carlos.solano@uv.es (C.S.); 2INCLIVA, Biomedical Research Institute, 46017 Valencia, Spain; 3Hematology Division, Hospital Morales Meseguer, 30100 Murcia, Spain; inmheras@um.es (I.H.); cris_aroca@hotmail.com (C.A.); evasolerespejo@gmail.com (E.S.-E.); 4Infectious Disease Division, Hospital Clinic, 08193 Barcelona, Spain; tfaiello@recerca.clinic.cat (T.F.A.); carolgv75@hotmail.com (C.G.-V.); 5Hematology Division, Hospital de la Santa Creu i Sant Pau, 08193 Barcelona, Spain; igarciaca@santpau.cat (I.G.-C.); aarrufat@santpau.cat (A.A.); rmartino@santpau.cat (R.M.); 6Hematology Department, University Hospital of Salamanca (HUS/IBSAL), CIBERONC and Cancer Research Institute of Salamanca-IBMCC (USAL-CSIC), 37007 Salamanca, Spain; lvazlo@usal.es (L.V.); lucialopezcorral@usal.es (L.L.-C.); aavendano@saludcastillayleon.es (A.A.-P.); lhmedina@saludcastillayleon.es (L.H.-M.); 7Hematology Division, Hospital Ramon y Cajal, 28029 Madrid, Spain; jljimenez@salud.madrid.org (J.L.-J.); jvalentingg@gmail.com (V.G.-G.); 8Hematology Division, Hospital Universitario y Politécnico La Fe, 46017 Valencia, Spain; pmachorao@gmail.com (P.C.); guerreiro_manuel@hotmail.com (M.G.); 9Hematology Division, Hospital Universitario Virgen Macarena, 41092 Sevilla, Spain; elena.arellano@juntadeandalucia.es; 10Hematology Division, Hospital de Fuenlabrada, 28029 Madrid, Spain; cgsantillana@salud.madrid.org; 11Hematology Division, Hospital Universitario Infanta Leonor, 28029 Madrid, Spain; jahr_jahr2006@yahoo.es; 12Institute of Experimental and Clinical Pharmacology and Toxicology, Center for Brain, Behavior and Metabolism (CBBM), University of Lübeck, 23562 Lübeck, Germany; 13Microbiology Service, Hospital Clínico Universitario de Valencia, 46010 Valencia, Spain; david.navarro@uv.es; 14Department of Medicine, School of Medicine. University of Valencia, 46010 Valencia, Spain; 15Hematopoietic Stem Cell Transplantation and Cell Therapy Group (GETH-TC) Office, 28029 Madrid, Spain; secretaria@geth.es

**Keywords:** remdesivir, nirmatrelvir/ritonavir, molnupiravir SARS-CoV-2 vaccines, Omicron, respiratory virus, hematological malignancies, allogeneic stem cell transplantation, autologous stem cell transplantation, COVID-19, mRNA vaccine, immunocompromised patients

## Abstract

Background: Scarce data exist that analyze the outcomes of hematological patients with SARS-CoV-2 infection during the Omicron variant period who received treatment with remdesivir or nirmatrelvir/ritonavir. Methods: This study aims to address this issue by using a retrospective observational registry, created by the Spanish Hematopoietic Stem Cell Transplantation and Cell Therapy Group, spanning from 27 December 2021 to 30 April 2023. Results: This study included 466 patients, 243 (52%) who were treated with remdesivir and 223 (48%) with nirmatrelvir/ritonavir. Nirmatrelvir/ritonavir was primarily used for mild cases, resulting in a lower COVID-19-related mortality rate (1.3%), while remdesivir was preferred for moderate to severe cases (40%), exhibiting a higher mortality rate (9%). A multivariate analysis in the remdesivir cohort showed that male gender (odds ratio (OR) 0.35, *p* = 0.042) correlated with a lower mortality risk, while corticosteroid use (OR 9.4, *p* < 0.001) and co-infection (OR 2.8, *p* = 0.047) were linked to a higher mortality risk. Prolonged virus shedding was common, with 52% of patients shedding the virus for more than 25 days. In patients treated with remdesivir, factors associated with prolonged shedding included B-cell malignancy as well as underlying disease, severe disease, a later onset of and shorter duration of remdesivir treatment and a higher baseline viral load. Nirmatrelvir/ritonavir demonstrated a comparable safety profile to remdesivir, despite a higher risk of drug interactions. Conclusions: Nirmatrelvir/ritonavir proved to be a safe and effective option for treating mild cases in the outpatient setting, while remdesivir was preferred for severe cases, where corticosteroids and co-infection significantly predicted worse outcomes. Despite antiviral therapy, prolonged shedding remains a matter of concern.

## 1. Introduction

The coronavirus disease 2019 (COVID-19) pandemic has revealed the heightened vulnerability of immunocompromised patients, especially those with hematological disorders, leading to increased morbidity and mortality compared to healthier individuals or patients with solid tumors [1,2]. However, the advancements in medical interventions, including supportive care, vaccination and antiviral drugs, have significantly decreased COVID-19 mortality rates from the first waves (>25%) [2,3] to the Omicron variant of concern (VOC) period (<2%) [4,5,6]. Interestingly, this positive trend was observed even before the introduction of SARS-CoV-2 vaccines and antiviral drugs in hematopoietic stem cell transplant recipients [7], also supporting the contribution of less severe emergent SARS-CoV-2 VOCs.

Direct-acting antiviral drugs such as remdesivir [8], VV116 [9] azvudine [10], nirmatrelvir/ritonavir [11] and, to a lesser extent, molnupiravir [12], have proven to reduce hospitalization, severe disease and death in the general population. Unlike vaccines and monoclonal antibodies, the antiviral activity of these drugs remains unaffected by the mutational evolution of SARS-CoV-2 VOCs [13], making them valuable tools against COVID-19, especially for immunosuppressed individuals who respond poorly to vaccination.

Despite a lack of clinical trials specifically evaluating the safety and efficacy of these antiviral drugs in immunosuppressed patients, national health authorities have approved their use and scientific societies have provided treatment recommendations for this vulnerable population [14]. For instance, nirmatrelvir/ritonavir is suggested for outpatient cases with asymptomatic or mild–moderate COVID-19, while remdesivir is preferred in moderate to critical cases, particularly for patients with pneumonia and oxygen requirements [14].

Understanding the safety and effectiveness of antiviral drugs in treating COVID-19 in immunocompromised patients is critical for optimizing treatment strategies and improving patient outcomes. To date, very few data exist on the safety and effectiveness of these antiviral drugs in treating hematological patients with SARS-CoV-2 infection or the underlying mechanisms that may influence their efficacy and tolerability [15,16,17,18].

In this context, the current multicenter, nationwide study assesses the outcomes of a large series of hematological patients with SARS-CoV-2 infection during the Omicron VOC period, who received treatment with remdesivir or nirmatrelvir/ritonavir in accordance with existing guidelines [14]. This study also investigates factors associated with poor outcomes and prolonged SARS-CoV-2 shedding. This research is based on a retrospective observational registry created by the Spanish Hematopoietic Stem Cell Transplantation and Cell Therapy Group (GETH-TC).

## 2. Patients and Methods

### 2.1. Study Population

In April 2023, the Infectious Complications Subcommittee (GRUCINI) of the GETH-TC launched a national retrospective multicenter registry that evaluated the outcome of Omicron VOC SARS-CoV-2 infection in immunocompromised patients with hematological diseases (HDs), including cell therapy recipients. The registry included consecutive HD patients with Omicron SARS-CoV-2 infection diagnosed either via PCR or rapid antigen tests. The status of all included patients was updated until 10 July 2023. The local Research Ethical Committee of the Hospital Clínico Universitario of Valencia approved the study protocol (reference code 35.21).

### 2.2. Inclusion Criteria, Variables of Interest and Clinical and Virological Monitoring

To assess the severity of epidemiologically defined Omicron VOC SARS-CoV-2 infections, the inclusion criteria were consecutive hematological patients with SARS-CoV-2 infection with clinical symptoms or asymptomatic cases who received antiviral therapy from 27 December 2021 to 30 May 2023. Variables of interest included demographics, base-line hematological disease characteristics, vaccination status before infection, clinical, virological and biological features of each SARS-CoV-2 infection episode (i.e., molecular or clinical resolution date); and data related to antiviral drugs such as schedule, duration, discontinuations, dose modifications and toxicities grade ≥3 according to the Common Toxicity Criteria (CTC). In this series, 53% of remdesivir-treated patients and 55% of those treated with nirmatrelvir/ritonavir were monitored on a weekly or biweekly basis until PCR negativity.

### 2.3. Definitions

Although we did not sequence SARS-CoV-2 strains in any case, the inference of the Omicron VOC was based on the Spanish sequencing epidemiological data, which started from 27 December 2021. The usage of corticosteroids was categorized into two main groups: individuals who were already receiving corticosteroid treatment at the time of SARS-CoV-2 infection and those who were administered corticosteroids as an anti-inflammatory therapy to manage severe COVID-19 cases. Complete vaccination schedules were defined as at least three vaccine doses (except COVID-19 Vaccine Janssen^®^, which requires at least 2 doses). Incomplete vaccination was considered when 2 or less doses were given before SARS-CoV-2 infection. SARS-CoV-2 infection was defined as molecular (PCR test) or antigenic evidence of SARS-CoV-2 infection. COVID-19 severity includes mild (no pneumonia nor oxygen support), moderate (pneumonia without oxygen support) and severe (pneumonia and oxygen support). Prolonged shedding was defined as persistent PCR positivity after 25 days from the first detection. The duration of COVID-19 was defined as the time from first detection until the first PCR negativity, or until symptom resolution in the case of no PCR monitoring. The length of SARS-CoV-2 shedding after treatment was calculated from the date of antiviral therapy until PCR negativity in those with available data. Co-infection was defined as a significant co-pathogen that was detected in concurrent nasopharyngeal or other body sites (including lungs, urine, blood or stools) that required antimicrobial intervention during SARS-CoV-2 infection and until its clinical and/or microbiological resolution. Co-infections also included the detection of community acquired respiratory viruses (CARVs) in the same sample as that of SARS-CoV-2. Regarding antiviral SARS-CoV-2 therapy, we considered a second antiviral course when a minimum break of at least 5 days was observed between courses to distinguish it from continuous and/or extended therapy. None of the patients in this current study received concomitant treatment with remdesivir plus nirmatrelvir/ritonavir.

### 2.4. Endpoints and Statistical Analysis

The primary objective of this study is to describe how and in what context antivirals have been prescribed, the clinical outcome of the treated patients and what factors are associated with these outcomes. Secondary end-points include the evaluation of prolonged shedding and to elucidate factors related with this condition, as well as to estimate the antiviral effect on viral burden through PCR cycle threshold (*Ct*) kinetics analysis. We also analyze the effect of other authorized therapies on SARS-CoV-2 infection outcomes, as well as safety issues related to antiviral drugs.

The main patient characteristics were reported using descriptive statistics on the total available information; medians and ranges were used for continuous variables, while absolute and percentage frequencies were used for categorical variables. For comparisons between antiviral cohorts, a Fisher exact test, Mann–Whitney’s U test or a median test were used when appropriate. Univariate and multivariate analyses of risk factors for SARS-CoV-2-related mortality were calculated using logistic regression models. A median test analysis to check the conditions that were potentially associated with prolonged shedding was carried out for each antiviral cohort in patients with available PCR monitoring data. A *p*-value < 0.05 was considered statistically significant. All *p*-values are two-sided. Analyses were performed using the statistical software SPSS v. 25(IBM SPSS Statistics, Armonk, New York, NY, USA).

## 3. Results

### 3.1. Patient Characteristics

The patient characteristics are summarized in Table 1. This study includes two cohorts: one treated with remdesivir (*n* = 243) and the other with nirmatrelvir/ritonavir (*n* = 223). Patients in both cohorts had a median age of 65 years (range 19–92), with a higher proportion of males in both groups. Baseline diseases varied among patients, with B-cell non-Hodgkin lymphoma (NHL) being the most common in both cohorts. The majority of patients in both cohorts experienced COVID-19 within 6 months of their last therapy. Anti-CD20 therapy was given in 31% and 40% of patients in the remdesivir and nirmatrelvir/ritonavir cohorts, respectively, whereas corticosteroids at the time of infection comprised over 20% of patients in both cohorts. The only significant differences in the baseline clinical characteristics between both cohorts were the vaccination status, with complete vaccination (at least three doses) being more common in the nirmatrelvir/ritonavir cohort (77% vs. 64%, *p* = 0.003) as well as a higher rate of previous pulmonary disease in the remdesivir cohort (14% vs. 7%, *p* = 0.05). The median follow-up for survivors after SARS-CoV-2 infection was 150 days (range 25–549).

### 3.2. Characteristics of SARS-CoV-2 Infection and Efficacy According to the Type of Antiviral Used

The clinical characteristics of SARS-CoV-2 infection are summarized in Table 2. The distribution of cases during the study period according to the antiviral drug are provided in Figure 1. The remdesivir cohort had a higher proportion of patients diagnosed using PCR compared to the nirmatrelvir/ritonavir cohort (90% vs. 60%, *p* < 0.001). The symptoms and severity of COVID-19 significantly varied between the cohorts, with the nirmatrelvir/ritonavir group showing a higher percentage of milder disease cases. Fever, respiratory symptoms, hospital admission, pneumonia and the need for oxygen support were significantly more prevalent in the remdesivir cohort (*p* < 0.001 for all comparisons). The duration of antiviral therapy also differed, with most nirmatrelvir/ritonavir patients (91%) receiving five days of treatment. However, 69 out of 466 (15%) (54 in the remdesivir group (22%) and 15 in the nirmatrelvir/ritonavir group, (7%)), received longer antiviral therapy. Co-infections were more common in the remdesivir cohort (*p* < 0.001). The nirmatrelvir/ritonavir group showed a higher rate of COVID-19 recovery at the last follow-up and a shorter median time to PCR negativity compared to the remdesivir cohort. Moreover, the nirmatrelvir/ritonavir cohort had a significantly lower COVID-19-related death rate (1.3% vs. 9%, *p* < 0.001) and fewer ICU admissions than the remdesivir group (0.5% vs. 6%, *p* < 0.001).

### 3.3. Univariate and Multivariate Analyses for Mortality

Table 3 presents the results of both the univariate and multivariate analyses that were conducted to identify risk factors for SARS-CoV-2-related mortality in the remdesivir cohort, consisting of 243 patients, out of which 21 patients experienced COVID-19 mortality. Such analyses were not performed in the nirmatrelvir/ritonavir group since such events were very uncommon (*n* = 3).

In the univariate analysis, variables associated with COVID-19 mortality were male gender, prior anti-CD20 therapy and timing from anti-CD20 therapy to COVID-19, pulmonary/cardiovascular risk factors, such as arterial hypertension, cardiomyopathy and pulmonary disease, the use of corticosteroids as COVID-19 therapy and co-infections. In the multivariate analysis, male gender was associated with a lower risk of mortality (odds ratio (OR) of 0.35, 95% confidence interval (C.I.) of 0.13–0.96, *p* = 0.042) whereas corticosteroid use showed a significant association with an increased risk of mortality (OR of 9.4, 95% C.I. of 2.9–30.2, *p* < 0.001), and co-infection (OR of 2.8, 95% C.I. of 1.01–7.7, *p* = 0.047) remained independently associated with a higher mortality risk. We also compared the COVID-19-related mortality in mild cases according to the antiviral drug used: only one case out of 209 in the N/R group died, compared to none in the 146 cases with mild disease who were treated with remdesivir (*p* = 0.99).

### 3.4. Cause of Death

The overall mortality rate in the remdesivir cohort was 21% (*n* = 50), whereas it was significantly lower in the nirmatrelvir/ritonavir cohort, with a mortality rate of 9% (*n* = 20) (*p* < 0.001). The median time to death was 53 days (range: 4–406) in the remdesivir group and 72 days (range: 18–296) in the nirmatrelvir/ritonavir group. Among the deceased patients, 12.4% (29 patients) in the remdesivir group and 7.7% (17 patients) in the nirmatrelvir/ritonavir group died from causes that were not related to COVID-19 (*p* < 0.001). Overall, the most common causes of death were baseline disease progression, accounting for 12% (29 cases) in the remdesivir group, and 4.5% (10 cases) in the nirmatrelvir/ritonavir group. Additionally, the nirmatrelvir/ritonavir group experienced nine cases of death due to other infectious complications after COVID-19 recovery, including one case of *pneumocystis jirovecii* pneumonia, three cases of pneumonia with an unknown origin, one case of invasive pulmonary fungal infection, three cases of septic shock and one case attributed to neurological deterioration caused by the JC virus.

### 3.5. Prolonged Shedding

SARS-CoV-2 PCR monitoring was available in 128 (53%) and 122 (55%) patients in the remdesivir and nirmatrelvir/ritonavir cohorts, respectively. Overall, the median duration of SARS-CoV-2 detection from the day of antiviral onset was 28.5 days (range 1–208) in the remdesivir cohort and 20 days (range 3–220) in the nirmatrelvir/ritonavir cohort (*p* = 0.004). Table 4 presents the median days of SARS-CoV-2 shedding, calculated from the day of antiviral drug onset according to different baseline factors in both cohorts.

In the remdesivir cohort, severe COVID-19 shed the virus for a longer duration compared to those with mild or moderate cases. Fever, pneumonia and the need for oxygen support were also linked to longer shedding, as was a longer timing from SARS-CoV-2 detection to antiviral onset, the duration of antiviral therapy in cases of persistent SARS-CoV-2 shedding, anti-CD20 treatment, the time from anti-CD20 treatment to COVID-19, B-cell NHL or chronic lymphocytic leukemia (CLL) diagnosis and a PCR *Ct* value of <20. Patients receiving corticosteroids for COVID-19 therapy had a longer viral shedding period, whereas those with incomplete vaccination showed shorter shedding duration. In the nirmatrelvir/ritonavir cohort, we did not find any condition associated with prolonged shedding.

Among the total patient population (*n* = 466), 22 patients (4%) required a second course of antiviral therapy due to the persistence of SARS-CoV-2 and/or worsening symptoms. In the remdesivir group, 12 patients (5%) needed a second course of treatment. Among these 12 patients, 10 received a repeat of remdesivir course, while two transitioned to nirmatrelvir/ritonavir therapy. In the nirmatrelvir/ritonavir group, another 11 patients (5%) received a second course of antiviral therapy, of whom four repeated a nirmatrelvir/ritonavir course, while six patients switched to remdesivir and one to molnupiravir. When we looked at those who suffered from prolonged shedding (80 out of 128 (63%) in the remdesivir group and 49 out of 122 (40%) in the nirmatrelvir/ritonavir group), we observed that a second antiviral course was given to seven out of eighty (9%) in the remdesivir groups and to five out of forty-nine (10%) in the nirmatrelvir/ritonavir group. Patients who received a second course showed a non-significant higher mortality (three out of twenty-three, 13%) as compared to those who only received one course (21 out of 443, 5%) (*p* = 0.108).

### 3.6. Antiviral Effect According to the Antiviral Drug

To assess the antiviral effect, we conducted a comparative analysis of the kinetics of the PCR *Ct* values in patients with mild COVID-19 severity. We specifically focused on individuals who had recorded *Ct* data at the time of initial SARS-CoV-2 detection and again after 2 weeks, if they had received antiviral drugs before the second PCR test. We categorized them based on the antiviral drug they received: remdesivir (*n* = 32 patients) or nirmatrelvir/ritonavir (*n* = 34 patients). For samples that tested negative at two weeks, we assigned a *Ct* value of 45 for representative purposes. In both groups, the median PCR *Ct* values increased after antiviral therapy (see Figure 2), with higher median *Ct* values in the nirmatrelvir/ritonavir group (*p* = 0.049). In the nirmatrelvir/ritonavir group, most patients achieved PCR negativity (29 out of 34 patients, 85%) compared to those in the remdesivir group (21 out of 32 patients, 65%) at the two-week mark, (*p* = 0.08). Eleven out of thirty-two patients (34%) from the remdesivir group had rebound infection and the *Ct* values remained the same (*n* = 3) or even decreased (*n* = 8) after 2 weeks. In the nirmatrelvir/ritonavir group, this phenomenon occurred in only five out of thirty four patients (15%) (two with the same *Ct* value and three with a lower *Ct*) (*p* = 0.09).

### 3.7. Tolerability of Antiviral Drugs

In Table 5, the tolerability and safety data of two antiviral treatments, remdesivir and nirmatrelvir/ritonavir, are compared. The data show that the risk of drug–drug interactions was significantly higher in the nirmatrelvir/ritonavir group compared to the remdesivir group (*p* = 0.001). Specifically, there were more cases of baseline treatment modifications (interruption or dose modification of baseline drugs), antiviral dose modifications and early antiviral discontinuations in the nirmatrelvir/ritonavir group. However, when it came to adverse events of grade ≥3, there was no significant difference between the two groups, with only one patient (0.5%) in the remdesivir group and three patients (1.3%) in the nirmatrelvir/ritonavir group experiencing such events (*p* = 0.3).

## 4. Discussion

We present herein a large cohort of hematological patients treated with remdesivir or nirmatrelvir/ritonavir during the Omicron SARS-CoV-2 period. Nirmatrelvir/ritonavir was primarily administered to mild cases (94%), resulting in a lower SARS-CoV-2 mortality rate (1.3%), while remdesivir was more frequently used for moderate to severe cases (40%) with a higher mortality rate (9%). Multivariate analysis in the remdesivir cohort revealed that male gender correlated with a lower mortality risk, while corticosteroid use as COVID-19 therapy and co-infection were linked to a higher mortality risk. Remdesivir-treated patients (with more severe cases) showed significantly longer SARS-CoV-2 shedding from the antiviral onset day compared to the nirmatrelvir/ritonavir group (28.5 days vs. 20 days). Additionally, in the remdesivir cohort, longer viral shedding was associated with the severity of COVID-19, baseline immunosuppression factors and the timing from illness onset to antiviral therapy. Conversely, the nirmatrelvir/ritonavir group displayed higher PCR *Ct* values at two weeks after treatment compared to the remdesivir group in cases with mild COVID-19, albeit with a higher risk of drug–drug interactions. Importantly, no significant difference in the incidence of grade ≥3 adverse events was observed between the two groups, indicating a similar safety profile for both antiviral drugs in this patient population.

This real-life retrospective study yielded several noteworthy observations. Firstly, the use of each antiviral closely adhered to the current recommendations [14]. Specifically, nirmatrelvir/ritonavir was mainly employed for mild infections in the outpatient setting, while remdesivir was of choice for severe cases during hospital admission. This adherence to guidelines potentially explains the significantly higher COVID-19-related mortality in the remdesivir group. Secondly, the study revealed a higher non-COVID-19 mortality within the remdesivir group, primarily attributed to a relapse or progression of baseline HDs. This observation suggests that this cohort may comprise frailer patients with aggressive and/or uncontrolled HDs at the time of contracting COVID-19. Furthermore, it is likely that this group had a higher prevalence of comorbidities, particularly pulmonary conditions, and the risk of drug–drug interactions with their baseline medication may have favored the use of remdesivir in this group. Such circumstances create challenges when attempting to compare the outcomes between antiviral treatment groups with different severity-based recommendations, making it potentially unfeasible [19].

Similar to prior retrospective reports from the Omicron period [15], we reported an overall COVID-19-related mortality of 5%. However, it should be noted that the current series mainly focused on symptomatic treated patients, disregarding non-treated and asymptomatic cases, which likely represent more than 50% of SARS-CoV-2 infection cases in prospective surveys [4,5,6]. Thus, it is plausible that the true COVID-19-related mortality could currently be lower than 2% in this vulnerable population in the current period.

In our analysis, we found three factors that were associated with mortality in the remdesivir group. Surprisingly, male gender had a protective effect against mortality, contrary to prior reports on the pre-Omicron VOC, suggesting a more severe disease course for males [20,21,22,23]. While we did not observe significant differences in the distribution of comorbidities between genders, as previously observed, it is possible that comorbidities may impact COVID-19 death more in women [24]. Remarkably, mortality was primarily influenced by the use of corticosteroids and co-infections during COVID-19. These factors were closely linked to each other and associated with higher mortality rates in critically ill COVID-19 patients, particularly with the use of dexamethasone [25]. In our series, the use of corticosteroids to manage COVID-19 was also closely linked to co-infections, which have been also associated with higher mortality in COVID-19 patients [26,27]. Still, the use of corticosteroids during infections in immunosuppressed patients is a matter of concern. Prior experience with the use of corticosteroids at the time of CARV infection in stem cell transplant recipients has been consistently associated with a higher risk of lower respiratory tract progression and mortality [28,29,30,31,32,33]. In contrast, based on the RECOVERY study in non-immunosuppressed individuals [34], many scientific societies have made high-grade recommendations on the use of dexamethasone in hematological malignancy patients with severe COVID-19, as happened with SARS-CoV-2 vaccine recommendations [14]. However, we already know from real-life experiences that caution is needed when extrapolating results from non-immunosuppressed individuals to immunosuppressed patients, as vaccine responses were significantly lower in the latter group, and a three-dose program of mRNA vaccines is now recommended [14]. Although one might argue that corticosteroids in our study were administered in severe cases, conceivably explaining the higher mortality, recent experience from the EPICOVIDEHA registry also supports an increased risk of mortality in hematological patients receiving corticosteroids, as shown in a propensity-score-matched analysis [35]. The limited inflammatory response in hematological patients during COVID-19, due to their profound cellular and humoral immunosuppression, raises serious concerns regarding corticosteroid use, facilitating co-infections in immunocompromised patients whilst managing COVID-19. Thus, it appears to be suitable to perform prospective randomized clinical trials before recommending their use in this setting.

Another differential aspect of COVID-19 in hematological patients as compared to the general population is the well-characterized prolonged virus shedding. While healthy individuals often shed non-viable virus fragments for an extended period, hematological patients may continue to shed viable SARS-CoV-2 for several months after the initial infection [36]. In our series, 52% of patients (149 out of 250) who were monitored through PCR until negativity showed a virus shedding for more than 25 days from the first detection. Prolonged viral shedding is the result of both a baseline compromised B- or T-cell function and the COVID-19 severity, along with a high viral burden. In this sense, we confirmed in the remdesivir group that conditions related to impaired B- and T-cell function (i.e., anti-CD20 treatment, the time from anti-CD20 treatment to COVID-19, B-cell NHL or CLL diagnosis and corticosteroid use for COVID-19) were significantly associated with prolonged virus shedding, in line with a prior report [37]. Importantly, the COVID-19 severity was also an important cause of prolonged shedding in our series, supported not only by the severe category but also by the development of symptoms such as fever, pneumonia and oxygen requirements. The fact that we only identified conditions that were associated with prolonged viral shedding in the remdesivir group supports this relation, since most severe cases were included in this group. In fact, the remdesivir cohort showed significantly longer SARS-CoV-2 shedding compared to the nirmatrelvir/ritonavir group (28.5 days vs. 20 days). A third condition associated with this phenomenon, and obviously linked to the disease severity, is the viral load, indirectly measured using the PCR *Ct* values, with a cut-off of 20 in our series. Prior studies have shown that severe cases exhibit an average viral load approximately 60 times greater than that of non-severe cases, indicating a potential correlation between higher viral loads and severe clinical outcomes [38]. However, early antiviral treatment can substantially reduce the duration of viral replication and accelerate the clearance of the virus [39]. In this sense, we also confirm in the remdesivir cohort that a period of less than 5 days from illness onset to antiviral therapy was significantly correlated with a shorter duration of SARS-CoV-2 shedding, supporting the idea that early therapy may contribute to improved outcomes [38,39,40,41]. Lastly, monitoring the dynamic change in the PCR *Ct* value can partially evaluate antiviral effectiveness [42]. Given that severe COVID-19 was over-represented in the remdesivir group, we restricted this analysis to mild COVID-19 cases in both groups, and we observed a higher antiviral effect in the nirmatrelvir/ritonavir cohort and a trend towards a shorter time to negativity. In vitro models have shown that both molecules have equipotent antiviral activity against the ancestral virus and the VOCs Alpha, Beta, Gamma, Delta and Omicron [13]. However, future research comparing the antiviral potency of each compound in vivo in this setting is warranted.

Finally, this study provides valuable insights into the safety of antiviral drugs in treating COVID-19 in immunocompromised patients with hematological diseases. As expected, nirmatrelvir/ritonavir was significantly associated with higher drug–drug interactions, although severe adverse events did not differ between the groups.

This study relies on historical data and observational analysis, potentially introducing limitations and biases. To substantiate these findings and establish stronger evidence for the efficacy of particular treatments in hematological patients during the Omicron variant period, further research and prospective studies are warranted.

## 5. Conclusions

In line with the current guidelines, remdesivir was the treatment of choice for severe cases, and mortality in this group was mainly associated with the use of corticosteroids and co-infections. Nirmatrelvir/ritonavir appears to be safe and effective in reducing SARS-CoV-2 shedding, having been a preferable option in the outpatient setting. Prolonged shedding was very common and closely linked to immunosuppression status, COVID-19 severity, viral burden and the timing from symptoms to antiviral onset.

## Figures and Tables

**Figure 1 viruses-15-02066-f001:**
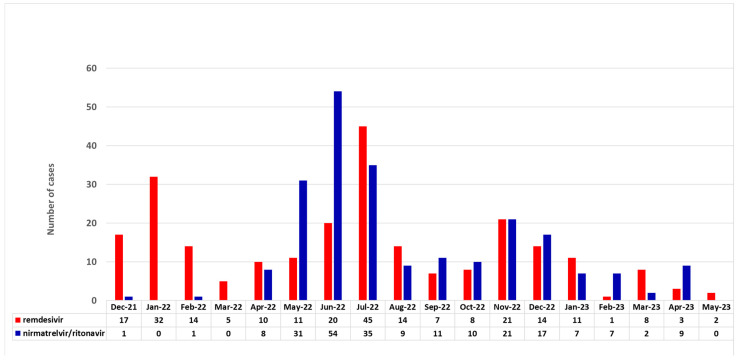
Distribution of antiviral use over the study period.

**Figure 2 viruses-15-02066-f002:**
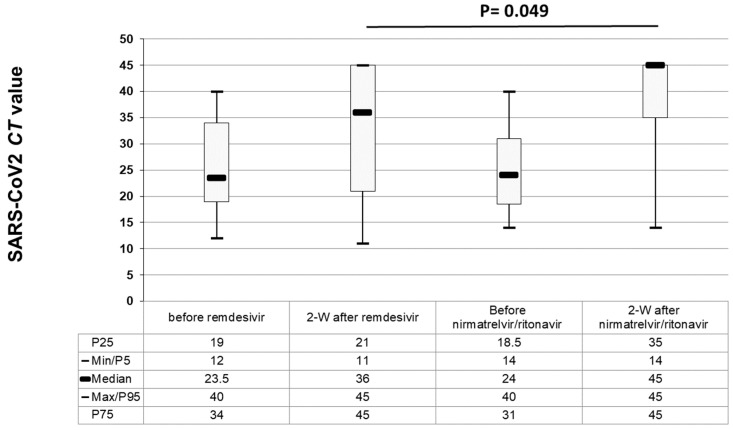
Cycle threshold (*Ct*) kinetics before and two weeks after antiviral treatment, according to the antiviral drug in 66 patients with mild COVID-19 (32 treated with remdesivir and 34 with nirmatrelvir/ritonavir).

**Table 1 viruses-15-02066-t001:** Patients’ characteristics.

Characteristics	Remdesivir Cohort (*n* = 243)	Nirmatrelvir/Ritonavir Cohort (*n* = 223)	*p* Value
**Age (years), median (range)**	64 (19–92)	65 (19–90)	0.4
• 0–40 years, *n* (%)	18 (8)	19 (9)	0.28
• 41–60 years, *n* (%)	86 (35)	66 (30)	
• 61–70 years, *n* (%)	49 (20)	60 (27)	
• >71 years, *n* (%)	90 (37)	78 (35)	
**Male, *n* (%)**	150 (62)	131 (59)	0.63
**Baseline disease, *n* (%)**			0.34
• AML	41 (17)	26 (12)	
• ALL	9 (4)	6 (3)	
• MDS	21 (9)	13 (6)	
• CMPD	8 (3)	7 (3)	
• B-cell NHL	78 (32)	88 (40)	
• T-cell NHL	7 (3)	2 (1)	
• CLL	20 (8)	12 (5)	
• Plasmatic cell disorder	38 (16)	57 (25.5)	
• HD	12 (5)	11 (5)	
• AA or others	6 (2)	1 (0.5)	
**Time from last therapy to COVID-19**			0.61
• <6 months	182 (75)	158 (71)	
• 6–12 months	19 (8)	20 (9)	
• >12 months or not treated	42 (17)	45 (20)	
**Anti-CD 20 and time from to COVID-19, *n* (%)**	74 (31)	88 (40)	0.13
• <6 months	56 (24)	68 (31)	
• 6–12 months	6 (3)	3 (1.5)	
• >12 months or not treated	12 (5)	17 (8)	
**Cell therapy, *n* (%)**	94 (39)	81 (36)	0.63
**- ASCT, *n* (%)**	36 (15)	41 (18)	
**- Allo-SCT donor, *n* (%)**	46 (19)	25 (11)	
• HLA identical sibling	21 (9)	8 (3.6)	
• Unrelated Donor	12 (5)	9 (4)	
• Haplo-identical family donor	12 (5)	8 (3.6)	
• UCBT	1 (0.5)	0	
**- CAR-T type, *n* (%)**	12 (5)	15 (7)	
• Axi-cell	6	6	
• Tisa-cell	1	2	
• Anti-BCMA	1	1	
• ARI-001 (anti-CD19)	4	5	
**Corticosteroids at the time of COVID-19, *n* (%)**	68 (28)	48 (22)	0.1
**Number of vaccine doses, *n* (%)**			0.017
• 0, *n* (%)	22 (9)	14 (6)	
• 1, *n* (%)	8 (3)	8 (4)	
• 2, *n* (%)	56 (23)	29 (13)	
• 3, *n* (%)	87 (36)	92 (41)	
• >3, *n* (%)	70 (29)	80 (36)	
**Vaccination status ©, *n* (%)**			0.003
• Incomplete	86 (36)	51 (23)	
• Complete	157 (64)	172 (77)	
**Tixagevimab/cilgavimab pre-exposure prophylaxis *n* (%)**	4 (1.6)	8 (3.6)	0.25
**Pulmonary/cardiovascular RF, *n* (%)**			
• Active smoking	21 (9)	21 (9)	0.87
• Arterial hypertension	105 (43)	79 (35)	0.2
• Cardiomyopathy	49 (20)	30 (14)	0.14
• Pulmonary disease	33 (14)	15 (7)	0.05
**Overall mortality, *n* (%)**	50 (21)	20 (9)	<0.001
**Median F/U after COVID-19, days (range)**	119 (3–549)	136 (5–539)	0.4

Abbreviations: AML, acute myeloid leukemia; ALL, acute lymphoblastic leukemia; MDS, myelodysplastic syndrome; cMPD, chronic myeloproliferative disease; NHL, non-Hodgkin lymphoma; CLL, chronic lymphocytic leukemia; HD, Hodgkin lymphoma; AA, aplastic anemia; ASCT, autologous hematopoietic stem cell transplantation; allo-SCT, allogeneic hematopoietic stem cell transplantation; UCBT, umbilical cord blood transplantation; CAR-T, chimeric antigen receptor T-cell therapy; BCMA, B-cell maturation antigen; ARI-001, academic CART-T; F/U, follow-up. © Complete vaccination was defined as at least 3 vaccine doses, whereas incomplete include 2 or less.

**Table 2 viruses-15-02066-t002:** Clinical and laboratory characteristics of SARS-CoV-2 infection.

Characteristics	Remdesivir Cohort (*n* = 243)	Nirmatrelvir/Ritonavir Cohort (*n* = 223)	*p* Value
**Diagnostic test, *n* (%)**			<0.001
• PCR	218 (90)	135 (60)	
• Antigen-test-based	25 (10)	88 (40)	
**Asymptomatic, *n* (%)**	2 (1)	6 (3)	0.1
**Fever, *n* (%)**	127 (52)	71 (32)	<0.001
**Respiratory symptoms, *n* (%)**	172 (71)	107 (48)	<0.001
**Pneumonia, *n* (%)**	97 (40)	14 (6)	<0.001
**COVID-19-related hospital admission, *n* (%)**	118 (49)	19 (9)	<0.001
**Oxygen support, *n* (%)**	69 (28)	5 (2)	<0.001
**COVID-19 severity, *n* (%)**			<0.001
• Mild	146 (60)	209 (94)	
• Moderate	28 (12)	9 (4)	
• Severe	69 (28)	5 (2)	
**Median time from dx to ATV therapy, days (range)**	1 (0–57)	1 (0–140)	
• <5 days, *n* (%)	190 (78)	203 (91)	<0.001
• ≥5 days, *n* (%)	53 (22)	20 (9)	<0.001
**Median duration of ATV therapy, days (range)**	5 (1–19)	5 (2–15)	
• <5 days, *n* (%)	89 (37)	4 (2)	<0.001
• 5 days, *n* (%)	100 (41)	204 (91)	<0.001
• >5 days, *n* (%)	54 (22)	15 (7)	<0.001
**Co-infection, *n* (%)**	46 (19)	9 (4)	<0.001
• Bacterial	35 (14)	5 (2)	
• viral	6 (3)	3 (1.5)	
• Fungal	5 (2)	1 (0.5)	
**Other COVID-19 therapies, *n* (%)**			<0.001
• Sotrovimab	43 (19)	7 (3)	
• Tixagevimab/cilgavimab	4 (2)	0	
➢ Median onset after COVID-19, days (range)	6 (0–129)	62 (0–104)	<0.001
• Convalescent plasma	30 (12)	6 (3)	<0.001
• Corticosteroids as COVID-19 therapy	79 (32)	6 (3)	<0.001
➢ Median onset after COVID-19, days (range)	2 (-5–46)	8 (0–79)	<0.001
**Laboratory characteristics at the time of SARS-CoV-2 detection**			
(A) ANC < 0.5× 10^9^/L, *n*/evaluable (%)	18/144 (13)	10/96 (10)	0.2
(B) ALC < 0.5× 10^9^/L, *n*/evaluable (%)	50/144 (35)	30/96 (31)	0.35
• CRP > 8 IU/mL, *n*/evaluable, %)	76/142 (53)	50/90 (55)	0.6
• *Ct* value at diagnosis available, *n* (%)	142 (58)	104 (47)	0.2
• *Ct* value at diagnostic, median (range)	20 (0–40)	21 (9–40)	0.8
**Recovery from COVID-19 at last F/U (*n*/evaluable, %)**	163/234 (70)	170/173 (98)	<0.001
• **PCR monitoring (*n*/evaluable, %)**	128/243 (53)	122/223 (55)	0.3
• **PCR negativity at last F/U, *n* (%)**	132 (63)	125 (69)	0.45
• **Median time from diagnosis to negativity, days (range)**	31 (2–209)	22 (2–237)	<0.001
• **Prolonged shedding, *n*/evaluable (%)**	80/128 (63)	49/122 (40)	<0.001
• **COVID-19-related death, *n* (%)**	21 (9)	3 (1.3)	<0.001
**Median time from diagnosis to death (range)**	30 (4–111)	24 (18–77)	0.2
**ICU admission, *n* (%)**	14 (6)	1 (0.5)	0.005

Abbreviations:, PCR, real-time polymerase chain reaction; dx, diagnosis of COVID-19; ATV, antiviral drugs; ICU, intensive care unit admission; ANC, absolute neutrophil count; ALC, absolute lymphocyte count; CRP, C-reactive protein; PCR, polymerase chain reaction; *Ct*, PCR cycle threshold; F/U, follow-up; ICU, intensive care unit.

**Table 3 viruses-15-02066-t003:** Univariate and multivariate analysis of risk factors for SARS-CoV-2-related mortality in the remdesivir cohort.

Variables	Log. Regr. COVID-19 Mortality * (*n* = 21 Out of 243)			
	Univariate Analysis		Multivariate Analysis	
	OR (95% C.I.) % (95%C.I.)	P	OR (95% C.I.)	P
**Age (years),**				
• <41 years	1			
• 41–60 years	0.6 (0.06–6.2)	0.68		
• 61–70 years	1.5 (0.15–14.5)	0.7		
• >71 years	2.87 (0.35–23)	0.3		
**Male, *n* (%)**	0.41 (0.17–1.06)	0.069	0.35 (0.13–0.96)	0.042
**Baseline disease, *n* (%)**			ns	
• AML	1			
• ALL	-			
• MDS	1.02 (0.12–3.5)	0.8		
• CMPD	1.7 (0.14–21)	0.65		
• B-cell NHL	3.9 (0.83–18)	0.08		
• T-cell NHL	3.2 (0.25–41)	0.36		
• CLL	3.44 (0.52–22)	0.19		
• Plasmatic cell disorder	0.52 (0.04–6.06)	0.6		
• HD	-			
• AA or others	-			
**Time from last therapy to COVID-19**				
• <6 months	1			
• 6–12 months	2.08 (0.54–7.9)	0.28		
• >12 months or not treated	0.85 (0.23–3.1)	0.8		
**Prior anti-CD20 therapy**	3.44 (1.38–8.5)	0.008	ns	
**Time from anti-CD 20 to COVID-19, *n* (%)**			ns	
• No anti-CD20	1			
• <6 months	2.47 (0.87–6.9)	0.08		
• 6–12 months	17.6 (3.12–100)	0.001		
• >12 months or not treated	3.5 (0.6–18.5)	0.13		
**Corticosteroids at the time of COVID-19**	1.03 (0.38–2.7)	0.9		
**Procedure**				
• Non-SCT	0.88 (.27–3.2)	0.8		
• Allo-SCT	1			
• ASCT	1.2 (0.06–5.6)	0.8		
• CAR-T	1.83 (0.25–13)	0.54		
**Vaccination status©**				
• Complete	1			
• Incomplete	1.1 (0.45–2.85)	0.78		
**Pulmonary/cardiovascular risk factors, *n* (%)**				
• Active smoking	0.5 (0.06–3.9)	0.5		
• Arterial hypertension	2.8 (1.1–7.2)	0.032	ns	
• Cardiomyopathy	3.3 (1.3–8.5)	0.011	ns	
• Pulmonary disease	2.8 (1.01–7.8)	0.05	ns	
**Other COVID-19 therapies**				
• Convalescent plasma	0.33 (0.04–2.5)	0.29		
• Monoclonal antibodies	0.99 (0.31–3.1)	0.9		
• Corticosteroids	11 (3.55–33.8)	<0.0001	9.4 (2.9–30.2)	<0.001
**Co-infection**	4.59 (1.8–11.6)	0.001	2.8 (1.01–7.7)	0.047
**ATV onset**				
• <5 days	1			
• ≥5 days	1.48 (0.54–4.04)	0.43		
**Duration of remdesivir**				
• <5 days	1.13 (0.4–3.1)	0.8		
• 5 days	1			
• >5 days	1.17 (0.36–3.78)	0.78		
***Ct* value at diagnosis ***				
• ≥20	1			
• <20	1.35 (0.41–4.3)	0.61		
**CRP > 8 UI/mL ***	1.8 (0.5–5.2)	0.9		
**ALC < 0.5 × 10^9^/L ***	0.74 (0.13–3.9)	0.7		
**ANC < 0.5 × 10^9^/L ***	1.1 (0.13–10.2)	0.9		

Abbreviations: SCT, stem cell transplantation; ASCT, autologous hematopoietic stem cell transplantation; allo-SCT, allogeneic hematopoietic stem cell transplantation; AML, acute myeloid leukemia; ALL, acute lymphoblastic leukemia; MDS, myelodysplastic syndrome; cMPD, chronic myeloproliferative disease; NHL, non-Hodgkin lymphoma; CLL, chronic lymphocytic leukemia; AA, aplastic anemia; *Ct*, PCR cycle threshold; ANC, absolute neutrophil count; ALC, absolute lymphocyte count; CRP, C-reactive protein; ns, not significant. * These variables have not been tested since there is a lack of data in more than 40% of cases. © Complete vaccination was defined as at least 3 vaccine doses, whereas incomplete includes 2 or less.

**Table 4 viruses-15-02066-t004:** Median days of SARS-CoV-2 shedding, calculated from the day of antiviral drug onset according to different conditions.

	Remdesivir Cohort Ω SARS-CoV-2 Shedding			Nirmatrelvir/Ritonavir Cohort Ω SARS-CoV-2 Shedding		
Variable	Median Days (Range)	*n*	*p* Value	Median Days (Range)	*n*	*p* Value
**Age**			0.25			0.87
• <41 years	21 (2–32)	7		17 (6–95)	12	
• 41–60 years	30 (2–162)	55		20.5 (2–220)	40	
• 61–70 years	30 (1–208)	32		17.5 (2–132)	30	
• >71 years	25 (2–108)	34		21 (2–167)	40	
**SARS-CoV-2 infection severity**			0.013			NT
• Mild	27 (2–162)	91		20 (2–220)	116	
• Moderate	27.5 (3–108)	16		7.5 (2–25)	4	
• Severe	39 (1–208)	21		5 (2–8)	2	
**Fever**			0.035			0.83
• Yes	32 (1–208)	61		20 (2–122)	42	
• No	23 (2–162)	67		19.5 (2–220)	80	
**Corticosteroids at infection**			0.82			0.13
• Yes	31 (3–208)	39		14.5 (2–75)	32	
• No	28 (1–162)	89		21 (2–220)	90	
**Pneumonia**			0.022			NT
• Yes	34 (1–208)	37		7.5 (2–25)	6	
• No	27 (2–162)	91		20.5 (2–220)	116	
**Hospital admission**			0.15			0.88
• Yes	30 (1–117)	69		25 (2–52)	11	
• No	24 (2–208)	59		20 (2–220)	111	
**Oxygen requirement**			0.022			0.8
• Yes	39 (1–208)	21		5 (2–8)	2	
• No	26.5 (2–162)	106		20 (2–220)	120	
**Monoclonal antibodies**			0.055			0.82
• Yes	35 (7–162)	26		20.5 (2–132)	11	
• No	25 (1–208)	98		20 (2–220)	111	
**Convalescent plasma**			0.7			0.49
• Yes	30 (4–106)	24		25 (2–59)	5	
• No	28 (1–208)	104		19 (2–220)	117	
**Corticosteroids for COVID-19 therapy**			0.004			0.11
• Yes	37 (1–208)	31		2 (2–25)	3	
• No	26 (2–117)	97		20 (2–220)	119	
**Time from last therapy to COVID-19**			0.49			0.87
• <6 months	29 (1–162)	104		19.5 (2–167)	90	
• 6–12 months	29 (11–62)	9		18.5 (2–122)	12	
• >12 months or not treated	17 (3–208)	15		22.5 (2–220)	18	
**Anti-CD20**			0.015			0.34
• Yes	33 (1–162)	39		20 (2–220)	59	
• No	27 (2–208)	89		20 (2–167)	63	
**Time from anti-CD 20 to COVID-19**			0.027			0.69
• <6 months	34 (1–162)	32		20 (6–132)	49	
• 6–12 months	67.5 (62–73)	2		32 (2–63)	2	
• >12 months	29 (7–63)	5		16 (2–220)	8	
• Not treated	27 (2–208)	89		20 (2–167)	63	
**B-cell NHL/CLL vs. others**			0.005			0.61
• Yes	33 (1–162)	49		19.5 (2–220)	64	
• No	25 (2–208)	79		20.5 (2–167)	58	
***Ct* value at diagnosis ***			0.001			0.82
• ≥20	19 (2–108)	49		19 (2–90)	28	
• <20	38 (1–208)	47		21 (5–95)	34	
**ATV onset from the first positive PCR**			0.008			0.6
• <5 days	27.5 (2–208)	100		20 (2–167)	110	
• ≥5 days	31 (1–162)	28		20.5 (2–220)	12	
**Duration of ATV**			0.001			0.83
• <5 days	19 (2–89)	42		21 (20–21)	2	
• 5 days	33 (1–208)	52		19 (2–220)	108	
• >5 days	31.5 (4–108)	34		24 (4–132)	12	
**Co-infections**			0.34			0.63
• Yes	34 (2–208)	17		38 (13–63)	2	
• No	27 (1–162)	111		20 (2–220)	120	
**ANC < 0.5 × 10^9^/L**			0.4			0.25
• Yes	38 (2–108)	11		14 (6–51)	5	
• No	30 (1–117)	77		24 (2–122)	49	
**ALC < 0.5 × 10^9^/L**			0.1			0.81
• Yes	35 (2–117)	36		20.5 (2–122)	18	
• No	27 (1–90)	52		23 (2–95)	36	
**Vaccination status**			0.039			0.28
• Complete	31 (1–208)	82		20 (2–220)	93	
• Incomplete	24.5 (2–90)	46		18 (2–167)	29	

Abbreviations: NHL, non-Hodgkin lymphoma; CLL, chronic lymphocytic leukemia; *Ct*, PCR cycle threshold; ANC, absolute neutrophil count; ALC, absolute lymphocyte count; NT, not tested. Ω Median duration of SARS-CoV-2 detection estimated from the day of antiviral onset in the remdesivir cohort was 28.5 days (range 1–208), whereas in the nirmatrelvir/ritonavir cohort, it was 20 days (range 3–220) (*p* = 0.004). * These variables have not been tested since there is a lack of data in more than 40% of cases.

**Table 5 viruses-15-02066-t005:** Tolerability and safety data.

Characteristics	Remdesivir (*n* = 194)	Nirmatrelvir/Ritonavir (*n* = 223)	*p* Value
**Risk of drug–drug interactions, *n* (%)**			0.001
• Baseline treatment modification	1 (0.5)	5 (2.2)	
• ATV dose modification	1 (0.5)	9 (4)	
• Early ATV interruption	27 (14)	38 (17)	
• Treatment completed	165 (68)	120 (54)	
**Adverse events ≥ grade 3, *n* (%)**	1 (0.5)	3 (1.3)	0.3

Abbreviations: ATV, antiviral therapy.

## Data Availability

Data are available upon formal request by email to the Spanish hematopoietic transplant and cell therapy group (GETH-TC).

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
