# Peer review of "Remdesivir or Nirmatrelvir/Ritonavir Therapy for Omicron SARS-CoV-2 Infection in Hematological Patients and Cell Therapy Recipients"

_viruses, 2023, doi:10.3390/v15102066_

Round 1
Reviewer 1 Report
The study by Piñana et al. highlighted that remdesivir was the treatment of choice for severe cases, and mortality in this group was primarily associated with the use of corticosteroids and co-infections. Nirmatrelvir/ritonavir, on the other hand, was found to be safe and effective in reducing SARS-CoV-2 shedding, making it a preferable option in the outpatient setting. The study also highlighted that prolonged shedding of the virus was very common and closely linked to factors such as immunosuppression status, COVID-19 severity, viral burden, and timing from symptoms to antiviral onset. These findings emphasize the importance of considering these factors when managing patients with SARS-CoV-2 infection.
I recommend accepting this article after MINOR REVISIONS.
1. It's important to note that this study is based on retrospective data and observational analysis, which may have limitations and biases. Further research and prospective studies are needed to validate these findings and provide more robust evidence for the use of specific treatments in hematological patients with SARS-CoV-2 infection during the Omicron variant period.
2. Remdesivir, the intravenously administered version (version 1.0) of GS-441524, is the first FDA-approved agent for SARS-CoV-2 treatment. While, oral GS-441524 derivative (VV116; version 2.0, targeting highly conserved viral RdRp) could be considered as game-changers in treating COVID-19 because oral administration has the potential to maximize clinical benefits, including decreased duration of COVID-19 and reduced post-acute sequelae of SARS-CoV-2 infection, as well as limited side effects such as hepatic accumulation. For the benefits of the readers, in lines 67-69, please supply more relevant knowledge about “VV116”. “Direct acting antiviral drugs such as remdesivir, VV116 (DOI: 10.1016/j.ejmech.2023.115503), azvudine (Signal Transduct. Tar., 6 (2021), p. 414), nirmatrelvir/ritonavir, and to a lesser extent molnupiravir, have proven to reduce hospitalization, severe disease, and death in the general population”.
3. In line 58, “COVID-19” please change with "coronavirus disease 2019 (COVID-19)”.
4. References should be added.
Author Response
We apreciate very much the feedback of this reviewer.
As sugested:
- we have now include a limitations paragraph with main limitations.
- we have added VV116 and azvudine and the appropriated references
- We have included coronavirus disease 2019 (COVID-19)
Reviewer 2 Report
The article in question is devoted to the effect of a number of drugs against coronovirus on hematological patients and cell therapy recipients.
In my opinion, the article is written very well, the conclusions correspond to the data obtained. The experiments were performed at a high methodological level. In addition, the high practical significance of the work should be noted. In principle, the article can be published, but the authors apparently forgot to add a list of references. This should be fixed
Author Response
we appreciated the positive comments from this reviewer.
Regarding the reference list, the authors do not completely understand the reviewer comment since reference list is already available at the end of the typescript.